# Elevated Biomarkers of Inflammation and Vascular Dysfunction Are Associated with Distal Sensory Polyneuropathy in People with HIV

**DOI:** 10.3390/ijms25084245

**Published:** 2024-04-11

**Authors:** Mohammadsobhan Sheikh Andalibi, Jerel Adam Fields, Jennifer E. Iudicello, Monica M. Diaz, Bin Tang, Scott L. Letendre, Ronald J. Ellis

**Affiliations:** 1Department of Neurosciences, University of California San Diego, San Diego, CA 92093, USA; mandalibi@ucsd.edu; 2Department of Psychiatry, University of California San Diego, San Diego, CA 92093, USA; jafields@health.ucsd.edu (J.A.F.); jiudicello@health.ucsd.edu (J.E.I.);; 3HIV Neurobehavioral Research Program, University of California San Diego, San Diego, CA 92093, USA; 4Department of Neurology, Multiple Sclerosis/Neuroimmunology Division, University of North Carolina at Chapel Hill School of Medicine, 170 Manning Drive, Campus Box 7025, Chapel Hill, NC 27599, USA; monica.diaz@neurology.unc.edu; 5Division of Infectious Diseases and Global Public Health, Department of Medicine, University of California San Diego, San Diego, CA 92093, USA

**Keywords:** HIV, polyneuropathy, inflammatory biomarkers, adhesion molecules

## Abstract

Distal sensory polyneuropathy (DSP) is a disabling, chronic condition in people with HIV (PWH), even those with viral suppression of antiretroviral therapy (ART), and with a wide range of complications, such as reduced quality of life. Previous studies demonstrated that DSP is associated with inflammatory cytokines in PWH. Adhesion molecules, essential for normal vascular function, are perturbed in HIV and other conditions linked to DSP, but the link between adhesion molecules and DSP in PWH is unknown. This study aimed to determine whether DSP signs and symptoms were associated with a panel of plasma biomarkers of inflammation (d-dimer, sTNFRII, MCP-1, IL-6, IL-8, IP-10, sCD14) and vascular I integrity (ICAM-1, VCAM-1, uPAR, MMP-2, VEGF, uPAR, TIMP-1, TIMP-2) and differed between PWH and people without HIV (PWoH). A cross-sectional study was conducted among 143 participants (69 PWH and 74 PWoH) assessed by studies at the UC San Diego HIV Neurobehavioral Research Program. DSP signs and symptoms were clinically assessed for all participants. DSP was defined as two or more DSP signs: bilateral symmetrically reduced distal vibration, sharp sensation, and ankle reflexes. Participant-reported symptoms were neuropathic pain, paresthesias, and loss of sensation. Factor analyses reduced the dimensionality of the 15 biomarkers among all participants, yielding six factors. Logistic regression was used to assess the associations between biomarkers and DSP signs and symptoms, controlling for relevant demographic and clinical covariates. The 143 participants were 48.3% PWH, 47 (32.9%) women, and 47 (33.6%) Hispanics, with a mean age of 44.3 ± 12.9 years. Among PWH, the median (IQR) nadir and current CD4+ T-cells were 300 (178–448) and 643 (502–839), respectively. Participants with DSP were older but had similar distributions of gender and ethnicity to those without DSP. Multiple logistic regression showed that Factor 2 (sTNFRII and VCAM-1) and Factor 4 (MMP-2) were independently associated with DSP signs in both PWH and PWoH (OR [95% CI]: 5.45 [1.42–21.00], and 15.16 [1.07–215.22]), respectively. These findings suggest that inflammation and vascular integrity alterations may contribute to DSP pathogenesis in PWH, but not PWoH, possibly through endothelial dysfunction and axonal degeneration.

## 1. Introduction

Distal sensory polyneuropathy (DSP) is a common and debilitating condition that affects up to 50% of people with HIV (PWH) [1]. This type of peripheral neuropathy causes symptoms such as distal neuropathic pain (DNP), paresthesias, and loss of sensation, which can significantly impact the daily life and well-being of PWH [2]. DSP also heightens fall risk [3] and may diminish physical activity due to DNP-related discomfort, exacerbating comorbidities [3,4,5,6].

While inflammation is acknowledged as a key player in DSP pathogenesis in HIV-positive individuals, data postdating modern antiretroviral therapy are sparse, with inconsistent correlations between HIV biomarkers (CD4+ T lymphocytes and viral load) [1,2] and DSP incidence noted in contemporary studies [7]. Elevated levels of pro-inflammatory cytokines such as IL-1, IL-2 receptor-alpha, and tumor necrosis factor (TNF) have been associated with DSP in PWH [8]. Pro-inflammatory cytokines—namely IL-1, IL-2 receptor-alpha, and TNF—are known to be secreted by HIV-infected monocyte-macrophage lineage cells adjacent to sensory neurons in dorsal root ganglia, potentially explaining the sensory predominance of the HIV-associated DSP [7,9,10,11,12].

HIV DSP shares many characteristics with diabetic sensory polyneuropathy, which has been strongly associated with compromised microvascular blood flow and alterations in proteins involved in tissue remodeling [13,14,15]. HIV infection is associated with microvascular injury due to chronic inflammation and immune dysregulation, leading to endothelial cell dysfunction. This can result in ischemia and metabolic disturbances of the vasa vasorum that may contribute to nerve damage and dysfunction. Altered levels of tissue inhibitors of metalloproteinase 1 (TIMP-1) and matrix metalloproteinases (MMP) have been associated with diabetic and HIV-induced neuropathogenesis [16,17,18]. In this study, we examined the potential role of inflammation and microvascular compromise in DSP signs and symptoms in the context of HIV by measuring a panel of cytokines and other markers selected based on theoretical considerations and previous research as described above. Additionally, we assessed how these biomarkers would differentially relate to DSP in PWH and people without HIV (PWoH).

## 2. Results

### 2.1. Participants

This study comprised 143 participants, of whom 48.3% were PWH, 33.6% were Hispanic, and 67.1% were male, with a mean age of 44.3 ± 12.9 years. Participant characteristics by HIV serostatus are summarized in Table 1. PWH had significantly more males than PWoH (91.3% vs. 44.6%, *p* < 0.001). Still, serostatus groups did not differ in age, and proportion of ethnicity, history of metabolic syndrome, diabetes, and CVD risk. Lifetime methamphetamine use disorder was more frequent in PWH compared with PWoH (*p* < 0.001). PWH exhibited evidence of ART-induced immune recovery, as indicated by higher current CD4+ T lymphocyte counts (median = 643 cells/mm^3^) compared to nadir CD4 counts (median = 300 cells/mm^3^); 80.6% of PWH were virally suppressed. The frequency of DSP and its symptoms, including neuropathic pain, paresthesias, and loss of sensation, was significantly higher in PWH than in PWoH (*p* < 0.05).

### 2.2. Factor Analysis

Very weak to strong correlations between inflammatory and vascular integrity biomarkers were identified (Appendix A). Correlated biomarkers were grouped into six factors with factor analysis: MCP-1, TIMP-1, IP-10, IL-8, and VEGF were loaded in Factor 1; VCAM-1 and sTNFR-II in Factor 2; uPAR, d-dimer, and IL-6 in Factor 3; MMP-2 in Factor 4; TIMP-2 in Factor 5; MMP-7 in Factor 6.

### 2.3. Association of Plasma Biomarkers of Inflammation and Vascular Integrity with DSP Signs and Symptoms

Simple logistic regression analyses showed that HIV seropositivity, diabetes mellitus, older age, CVD risk, and Factors 2 (sTNFRII and VCAM-1) and 4 (MMP-2) were significantly associated with DSP (ps < 0.05, Appendix A). In addition, HIV status and Factor 2 were associated with DSP symptoms, including dysesthesia, paresthesia, and loss of sensation, as well as the presence of any of these symptoms. The variables associated with DSP signs or symptoms were included in multivariable analyses as covariates (i.e., age, gender, ethnicity, height, diabetes mellitus, metabolic syndrome, modified Framingham CVD risk score, current CD4+, HIV RNA, duration of HIV infection, current use and duration of cART, and history of lifetime alcohol, methamphetamine, and opioid use disorders). After adjusting for relevant sociodemographic and clinical covariates, multiple logistic regression indicated significant associations between Factor 4 and DSP (OR [95% CI]: 5.13 [1.21–21.86]) and Factor 2 and loss of sensation (2.08 [1.08–3.99], Figure 1). The interaction of HIV serostatus and factors as a predictor of the DSP and its symptoms was determined using multivariable regression. The results showed the interaction of HIV seropositivity and Factor 2 in predicting DSP and loss of sensation was significant. In stratified analysis by HIV serostatus, we observed that increased values of Factor 2 (sTNFRII and VCAM-1) were significantly associated with both DSP and loss of sensation in PWH (5.45 [1.42–21.00] and 2.30 [1.08–4.91], respectively), whereas increased Factor 4 (MMP-2) was associated with DSP signs in PWoH (15.16 [1.07–215.22]) (Appendix A). In addition, increased CVD risk and a history of methamphetamine use disorder were the other independent determinants of DNP and paresthesia in PWH, respectively (Appendix A).

### 2.4. Causal Mediation Analyses of Factor 2 (sTNFRII and VCAM) on the Relationships between HIV Serostatus and DSP Signs and Symptoms

We conducted mediation analyses to assess the interrelationship between HIV serostatus, DSP and its symptoms, and Factor 2. Our goal was to determine whether Factor 2 mediates the influence of HIV status on DSP symptoms, including dysesthesia, paresthesia, and loss of sensation. The results indicate that both the direct and indirect effects of HIV on DSP symptoms were significant via the mediator, Factor 2. Furthermore, Factor 2 accounted for 18.8–20.0% of the association between HIV and DSP symptoms (*p* < 0.05, Table 2).

## 3. Discussion

In middle-aged PWH and PWoH samples, we found that DSP signs correlated with higher levels of a factor loading on MMP-2. This enzyme degrades type IV collagen, the major structural component of basement membranes. MMP-2 is involved in vascular integrity, inflammation, angiogenesis, and neurogenesis [19]. Self-reported loss of sensation was associated with higher levels of a factor loading on VCAM and sTNFR-II. These findings were robust after consideration of covariables, such as diabetes. VCAM-1 is a marker of vascular dysfunction and regulates inflammation-associated vascular adhesion and the transendothelial migration of leukocytes, while sTNFR-II is a marker of inflammation. The association between HIV status and DSP signs and symptoms, including dysesthesia, paresthesia, and loss of sensation, was significant. Moreover, the factors associated with DSP signs and symptoms differed in PWH (Factor 2, VCAM-1, and sTNFRII) and PWoH (Factor 4, MMP-2), suggesting that the biological determinants of neuropathy signs and symptoms differ between people with and without HIV. sTNFR-II, linked to neurodegeneration, not only affects TNF-α activity but also shows a strong connection to the stage and advancement of HIV disease [20]. These features emphasize TNF-α’s influence on the immune system and DSP and suggest possibilities for treatment and monitoring HIV progression. In mediation analyses, we also showed that both direct and indirect effects of HIV on DSP symptoms were significant via the mediator, Factor 2 (VCAM/sTNFR-II) [21]. This is consistent with an underlying causal relationship between inflammation/vascular integrity and DSP symptoms. Since there was no significant association between Factor 4 and HIV serostatus, we did not run mediation analyses for Factor 4 (MMP-2).

The findings presented here are consistent with previous studies showing that perturbations in the interaction between inflammation and mitochondria may influence neuropathy symptoms in PWH. Expression levels of dynamin-related protein (DRP1), a key enzyme promoting mitochondrial fission, are altered in diseases characterized by chronic neuroinflammation, and it affects neuronal apoptosis, synaptic activity, and axonal integrity [22,23]. Administration of DRP1 inhibitors resulted in decreased HIV-associated neuropathy as well as TNF-induced mechanical hyperalgesia in rats [24]. DRP1 inhibition attenuated the expression of vascular adhesion molecules (ICAM-1 and VCAM-1) in the brains of a mouse model for septic encephalopathy [25]. This suggests a linkage between DRP1-related mitochondrial dysfunction and the adherence/transmigration of inflammatory cells across the disrupted blood–brain barrier (BBB) [26]. Our current findings show that VCAM and sTNFRII levels mediate DSP symptoms and are consistent with a role for mitochondrial fission in HIV-associated neuropathy. However, we did not observe any association between plasma ICAM-1 and DSP signs and symptoms in PWH, PWoH, and all study participants.

Our study is the first to systematically compare the relationship between vascular integrity markers and DSP in people with and without HIV. However, our findings corroborate several studies that reported elevated levels of inflammatory biomarkers and cytokines in pain-related conditions [27,28,29,30]. Soluble adhesion molecules (sICAM-1 and VCAM), among other markers, have been found to correlate with pain [31,32]. Measuring sVCAM-1 has shown a relationship with patients’ self-reported menstrual pain [31].

Cytokines also play a strong role in neuropathic pain. sTNFR-II, linked to neurodegeneration, not only affects TNF-α activity but also shows a strong connection to the stage and advancement of HIV disease [20]. Both functions emphasize its influence on controlling the immune system and evaluating the HIV-associated comorbidities, namely DSP, and suggesting possibilities for treatment and monitoring HIV progression. To our knowledge, this is the first study that investigated the linkage between sTNFRII and HIV-associated DSP; however, some other studies suggested the association between TNF-α and neuropathy. It has been reported that serum levels of inflammatory biomarkers, including C-reactive protein (CRP) and TNF-α, were higher in individuals with neuropathic pain than controls [33]. A comprehensive meta-analysis examining the relationship between serum TNF-α levels and diabetic peripheral neuropathy in patients with type 2 diabetes revealed increased TNF-α levels in patients with diabetic neuropathy compared to those without neuropathy and controls [34]. Furthermore, the recovery from neuropathic pain following peripheral nerve injury depends on the downregulation of IL-1 β and TNF- α responses [35].

Concordant with previous reports [29,36], in the current study, the frequency of DSP and its symptoms, including DNP, paresthesia, and loss of sensation, was much higher in virally suppressed PWH than in PWoH. This is consistent with DSP progression in older PWH, which aligns with the notion of an ongoing peripheral neurodegenerative process. CVDs are one of the major complications in virally suppressed PWH [37]. We found that CVD risk, according to the modified Framingham risk score, was independently associated with dysesthesia and paresthesia. However, no significant association was observed between metabolic syndrome and DSP, which is in accordance with the previous studies [38].

Interruptions in ART for PWH can lead to adverse outcomes, such as worsening of DSP symptoms and an increased risk of HIV disease progression due to decreased CD4 counts and nadir CD4 counts. Specifically, PWH who initiated CART after their CD4 counts dropped below 350 cells/μL were found to have a significantly higher prevalence of DSP compared to those who started CART before their CD4 counts fell below this threshold [2]. Moreover, some studies suggest that ART itself may be neurotoxic, as it has been associated with a higher prevalence of DSP even after controlling for other factors, including exposure to D-drugs [2]. However, further research is needed to fully understand the relationship between ART and DSP in PWH. The current study found that 87% of PWH were receiving ART, and no correlation was observed between the duration of ART and the presence or severity of DSP. However, a history of exposure to D-drugs was associated with an increased risk of sensory loss in PWH.

DSP is more common in the lower limbs than in the upper limbs. Research indicates that patients with additional causes of DSP more often experience upper limb sensory symptoms and findings, suggesting a higher prevalence of DSP in the lower extremities. In the current study, we also focused on bilateral distal vibratory, sharp, and touch loss in the legs and feet, and reduced ankle reflexes compared to knees. Although the clinical diagnosis of DSP was solely clinical, and any conductive or imaging studies, such as electromyography, intraepidermal nerve fiber density, and MRI, have not been performed for the participants, we have published previously on the correspondence of clinical evaluations of DSP and nerve conductive studies [39]. In addition, bilateral reflex loss (ankle jerk) is not common in most radiculopathies. However, we acknowledge that the signs might be overlapped between DSP and some radiculopathies, such as S1 radiculopathy [40].

Some limitations exist in our study. First, this was a cross-sectional study of DSP symptoms among PWH; therefore, further longitudinal studies are needed to define the trend of inflammatory biomarkers over time with the evolution of symptoms. Second, we could not figure out causal associations between inflammatory biomarker levels and DSP. Further, we did not measure IL-1, IL-2 receptor-alpha, and TNF, the biomarkers that have been previously observed to be linked with DSP, in this set of samples. We acknowledge the significance of these excluded cytokines in the context of DSP pathogenesis in PWH W and recommend designing a study to figure out the association between these inflammatory biomarkers and other plasma/CSF biomarkers with DSP in future projects. Lastly, there were few women in the PWH. Although the frequency of DSP did not differ between males and females, studies focusing on biomarkers specific to women with HIV and DSP are recommended.

The results of this study have significant implications for clinical practice. Symptoms are the main reason why patients seek medical attention. In the current study, we found that increased Factor 2 (sTNFRII and VCAM) contributes to the DSP symptoms, including paresthesia and loss of sensation, either by mediating the effect of HIV on DSP or by independently affecting the sensation loss. Of note, paresthesia is known as a predictor of prospective neuropathic pain in PWH with DSP [41]. DSP is associated with lower quality of life [42], reduced daily functioning [36], cognitive impairment [43], and balance problems [3]; preventing or alleviating it might improve these aspects of well-being. DSP constitutes a paramount complication among PWH. This study implies the plausible utility of interventions directed toward modulating inflammatory biomarkers, thereby attenuating the propensity for DSP in the HIV-infected population. It is also advisable to contemplate targeting inflammatory biomarkers and addressing endothelial dysfunction in PWH manifesting symptoms of DSP.

As for future research directions, it would be beneficial to investigate the mechanisms that link HIV, inflammatory biomarkers, and DSP. Understanding these interactions could shed light on the mechanisms underlying the association between HIV, inflammatory biomarkers, and DSP. It would be also worthwhile to examine the role of other inflammatory biomarkers and adhesion molecules in DSP development among HIV patients. This could broaden the understanding of DSP and lead to the discovery of new therapeutic targets.

## 4. Materials and Methods

### 4.1. Study Design and Participants

This cross-sectional observational study was conducted in a sample of 143 participants (69 PWH and 74 PWoH) assessed as part of NIH-funded studies conducted at the UC San Diego HIV Neurobehavioral Research Program (HNRP) from 2014 to 2020. We excluded participants with incomplete clinical evaluations, patients with active opportunistic infections or uncontrolled major psychiatric disorders, or neurological conditions such as epilepsy and schizophrenia. Comorbidities such as HCV infection and substance abuse were permitted. This study was performed according to the guidelines of the Declaration of Helsinki. All participants signed local IRB-approved written consents.

### 4.2. Clinical Evaluations

Participants underwent thorough standardized psychiatric and neuropsychological assessments, forming part of an extensive evaluation encompassing physical and neurological examinations. This evaluation also involved gathering standardized medical history and conducting routine clinical blood tests, such as measuring HIV RNA levels in plasma through the reverse transcription-polymerase chain reaction (RT-PCR), assessing CD4+ T-cell count via flow cytometry, and performing rapid plasma reagin tests.

#### Distal Sensory Polyneuropathy (DSP) Evaluations

Participants underwent standardized evaluations of DSP conducted by centrally trained clinicians, mid-level practitioners, and physicians. Evaluations included clinical examination for neuropathy signs: bilateral symmetric distal vibratory, sharp, and touch loss in the legs and feet, and reduced ankle reflexes compared to knees. DSP was defined as having ≥2 DSP signs [36]. DSP symptoms included DNP, paresthesias, and loss of sensation. Evaluations also included assessment of DNP, defined as burning, aching, or shooting symptoms in the distal legs and feet. To exclude non-DSP-related conditions, such as mononeuropathies and radiculopathies, we mandated the presence of symmetric, bilateral, and distal abnormalities with a stocking rather than a segmental pattern when considering both signs and symptoms. Notably, we have published previously on the correspondence of clinical evaluations of DSP and nerve conductive studies [39].

### 4.3. Laboratory Evaluations

HIV infection was diagnosed and confirmed by enzyme-linked immunosorbent assay with Western blot confirmation. HIV RNA in plasma was measured using commercial assays; viral suppression was defined as a level below the lower limit of quantitation of 50 copies/mL. Peripheral blood CD4+ T cell concentration was measured by flow cytometry. Plasma for biomarker assays was collected using EDTA vacuum tubes via standard phlebotomy procedures. Soluble levels of inflammatory biomarkers, including d-dimer, IL-6, IL-8, CXCL10/IP-10, CCL2/MCP-1, matrix metalloproteinase-2 (MMP-2), MMP-7, CD14, the receptor for tumor necrosis factor type II (sTNFRII), tissue inhibitor of metalloproteinase-1 (TIMP-1), TIMP-2, and biomarkers reflective of endothelial activation and/or vascular permeability (intercellular and vascular adhesion molecules-1 (ICAM-1, VCAM-1), vascular endothelial growth factor (VEGF), and uPAR) were measured using electrochemical luminescence immunoassays (MesoScale Discovery, MD, USA). Log10 transformation was used to reduce the skewness of biomarker distributions for parametric analysis. All assays were carried out in duplicate. To adjust for possible batch effects due to different measurement kits, concentrations of each inflammatory biomarker were regressed on plate number, and the residual values were used in all subsequent analyses.

### 4.4. HIV-Related Assessments and Potential Confounders

Current CD4+ T-cell count, nadir CD4+ count (self-report), viral suppression, plasma and CSF HIV viral load, AIDS status, duration of HIV infection, current use, and ART duration, and prior exposure to the D-drugs (Dideoxynucleoside analog reverse transcriptase inhibitors (nRTIs) including stavudine or didanosine) were measured and collected for all participants. Potential confounders known to be risk factors for DSP were considered and assessed. These included a history of diabetes mellitus (self-reported or antidiabetic medications) [44], metabolic syndrome according to AHA criteria [45], sex-specific cardiovascular disease (CVD) risk determined by the modified Framingham risk score [46,47], as well as anthropometric and demographic characteristics, such as age, gender, ethnicity, and height [36,44,48]. Information regarding the history of methamphetamine and other substance abuse and dependency was gathered using the Composite International Diagnostic Interview (CIDI version 2.1) [49,50]. This interview was structured according to DSM-IV criteria. Other Substance Use Disorders were identified as meeting specific criteria, including lifetime diagnoses of alcohol, cocaine, and opioid abuse or dependence.

### 4.5. Statistical Analyses

All statistical analyses were performed using SPSS version 28.0.1.1 (IBM SPSS Statistics) and JMP Pro 16.0.0 (SAS Institute Inc., Cary, NC, USA, 2021) software packages. Data were summarized as the mean ± SD or median [IQR] for normally or non-normally distributed variables, or N (%) for categorical variables. As appropriate, group comparisons were performed using the independent-samples *t*-test, the Mann–Whitney test, the Chi-squared test, or Fisher’s exact test.

Correlations of the fifteen biomarkers were evaluated using Pearson’s correlation coefficients (Appendix A), and factor analysis was conducted using Equamax rotation with a factor loading value of >0.45 to identify latent factors (i.e., sets of inflammatory biomarkers with mutual associations).

Simple logistic regression was performed to determine associations between demographic and clinical characteristics and outcomes (i.e., DSP signs, DNP, paresthesia, loss of sensation, and the presence of any DSP symptoms). Variables associated with outcome at *p* < 0.1 in univariable analysis were included in multiple logistic regression and retained as covariates if their association with outcome was significant at *p* < 0.05 after model selection with backward stepwise method. These analyses were performed among all participants and in subgroups stratified by HIV status.

Mediation analysis was conducted using the PROCESS extension package in SPSS to estimate the magnitude of the mediation effect (indirect effect [IE]) of Factor 2 on the relationship between HIV serostatus and DSP signs and symptoms (direct effect [DE]). The estimation of the mediating effect was conducted on the scale of odds ratios (OR), and 95% confidence intervals (CIs) were obtained using bootstrapping. The proportion mediated (PM) indicates the percentage of the mediation effect and is calculated using the formula PM = (TE − DE)/TE. TE represents the total effect, which decomposes into indirect effect (IE) and direct effect (DE), expressed as TE = IE + DE in the equation [51]. A two-sided *p* value < 0.05 was considered to be significant.

## Figures and Tables

**Figure 1 ijms-25-04245-f001:**
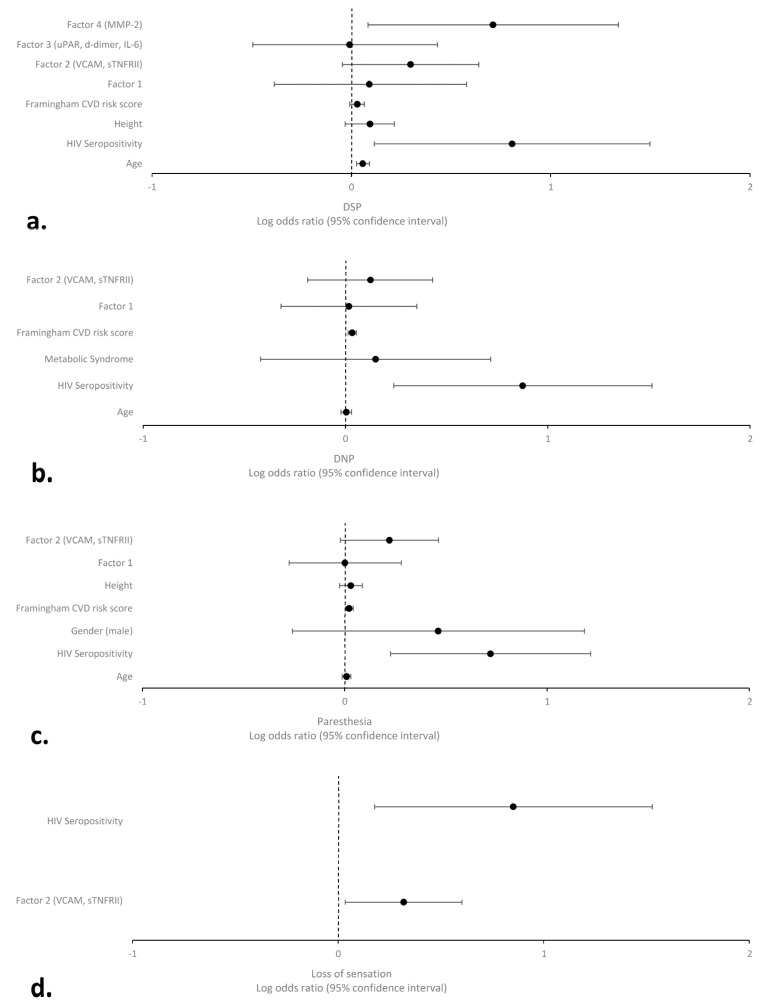
Log odds ratio for DSP (**a**), DNP (**b**), paresthesia (**c**), and loss of sensation (**d**) predictors. The model was adjusted for age, gender, ethnicity, height, diabetes mellitus, metabolic syndrome, modified Framingham CVD risk score, current CD4+, HIV RNA, duration of HIV infection, current use and duration of cART, and history of lifetime alcohol, methamphetamine, and opioid use disorders. Factor 1: MCP−1, TIMP−1, IP−10, IL−8, and VEGF. Abbreviations: DSP: distal sensory polyneuropathy; DNP: distal neuropathic pain.

**Table 1 ijms-25-04245-t001:** Comparison of demographic and clinical characteristics for PWH and PWoH participants.

	PWH n = 69	PWoH n = 74	All n = 143	*p*-Value
Age (year)	43.9 ± 15.3	44.7 ± 12.6	44.3 ± 12.9	0.74
Sex (male)	63 (91.3%)	33 (44.6%)	96 (67.1%)	**<0.001**
Ethnicity				
Non-Hispanic White	34 (50.0%)	36 (48.6%)	70 (49.3%)	0.32
Black	7 (10.3%)	15 (20.3%)	22 (15.5%)	
Hispanic	23 (33.8%)	18 (24.3%)	41 (28.9%)	
Other	4 (5.9%)	5 (6.8%)	9 (6.3%)	
Nadir CD4 * median (IQR)	300 (178–448)			
Current CD4 * median (IQR)	643 (502–839)			
Duration of infection (year) median (IQR)	8.7 (2.5–16.6)			
On ART	60 (87.0%)			
Duration of ART (month) median (IQR)	55.6 (20.6–142.9)			
Plasma HIV RNA ≤ 50 copies/mL	54 (80.6%)			
CSF HIV RNA ≤ 50 copies/mL	41 (97.6%)			
DSP variables				
Reduced vibration	13 (20.3%)	7 (10.3%)	20 (15.2%)	0.10
Reduced sharp sensation	5 (7.8%)	2 (3.0%)	7 (5.3%)	0.21
Reduced or absent ankle reflexes	19 (29.7%)	12 (17.6%)	31 (23.5%)	0.10
Pain	13 (20.3%)	3 (4.4%)	16 (12.1%)	**0.005**
Tingling	21 (32.8%)	5 (7.4%)	26 (19.7%)	**<0.001**
Loss of sensation	15 (23.4%)	2 (2.9%)	17 (12.9%)	**<0.001**
0–1 DSP signs ^	53 (82.8%)	64 (94.1%)	117 (88.6%)	**0.041**
≥2 DSP signs ^	11 (17.2%)	4 (5.9%)	15 (11.4%)	
Presence of any DSP symptoms	25 (39.1%)	8 (11.8%)	33 (25.0%)	**<0.001**
Concurrent medical conditions				
History of diabetes	3 (4.5%)	6 (8.5%)	9 (6.5%)	0.34
Metabolic syndrome	24 (34.8%)	27 (36.5%)	51 (35.7%)	0.83
Framingham CVD risk score				
Low risk	36 (58.1%)	44 (65.7%)	80 (62.0%)	0.37
Mid-high risk	26 (41.9%)	23 (34.3%)	49 (38.0%)	
Substance use disorders				
Lifetime alcohol use disorder	38 (60.3%)	28 (43.1%)	66 (51.6%)	0.051
Lifetime methamphethamine use disorder	40 (63.5%)	22 (33.8%)	62 (48.4%)	**<0.001**
Lifetime opioid use disorder	2 (3.2%)	3 (4.6%)	5 (3.9%)	0.67

* CD4+ T lymphocytes/uL. ^ DSP signs: reduced distal vibratory sensation, reduced distal pin sensation, and reduced or absent ankle reflexes. ART: anti-retroviral therapy; CSF: cerebrospinal fluid; DSP: distal sensory polyneuropathy; PWH: people with HIV; PWoH: people without HIV.

**Table 2 ijms-25-04245-t002:** Mediating effects of Factor 2 (sTNFRII and VCAM) on the association between HIV serostatus and risk of DSP signs and symptoms (N = 132).

Outcomes	Total Effect (95% CI)	Direct Effect (95% CI)	Mediation Effect (95%CI)	Proportion Mediated
DSP *	3.44 (0.62–22.26)	2.39 (0.68–8.40)	1.43 (0.91–2.67)	29.3
Presence of any Symptoms	5.07 (1.49–21.75)	3.66 (1.44–9.29)	1.38 (1.03–2.35)	20.0
DNP	5.64 (0.94–39.25)	4.40 (1.14–16.94)	1.28 (0.82–2.32)	14.4
Paresthesia (Tingling)	6.56 (1.56–34.12)	4.61 (1.55–13.65)	1.42 (1.01–2.51)	18.8
Loss of sensation	11.13 (1.53–114.4)	7.09 (1.50–33.44)	1.59 (1.03–3.42)	19.3

* DSP was defined as the presence of two or more of the following DSP signs: reduced, bilateral, symmetric, distal vibration, sharp sensation, and ankle reflexes. DSP: distal sensory polyneuropathy; DNP: distal neuropathic pain.

## Data Availability

The datasets generated and/or analyzed during the current study are available in the HNRP repository upon reasonable request.

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
