# Peer review of "Elevated Biomarkers of Inflammation and Vascular Dysfunction Are Associated with Distal Sensory Polyneuropathy in People with HIV"

_ijms, 2024, doi:10.3390/ijms25084245_

Round 1
Reviewer 1 Report
Comments and Suggestions for Authors
As I understand it, in effect this paper by and large compares aspects of what has been diagnosed as distal sensory polyneuropathy between 11 sufferers with the HIV-associated variety and 4 sufferers with other varieties of the neuropathy, however caused. If so, it seems to me that correct neurological categorisation of the disorders present is important. The diagnostic criteria described in the paper certainly indicate that the diagnostic sensory changes need to be symmetrical and distal in the lower limbs, but do not specify that their distribution be in a stocking rather than a segmental pattern, and do not consider the upper limbs where supporting evidence might be available. The bilateral reduction in ankle jerks relative to knee-jerks could be explained by lumbar-sacral disc protrusions at spinal level involving the S1 nerve roots bilaterally but sparing the L3,4 roots, with sensory change of the appropriate parts of both feet. Such disc protrusions are not rarities. Was their possible presence excluded?
Were nerve conduction studies carried out to support to the diagnostic categorisation?
Although you consider other possible causes of the peripheral nerve pathology such as diabetes, alcohol, and various non-therapeutic drugs, did you take steps to exclude other possible non-HIV causes such as familial varieties of peripheral neuropathy, industrial exposure to neurotoxins or certain therapeutic drugs that may have been prescribed in the past?
It seems to me that answers to the above questions might be helpful to readers in their assessing of your paper.
Author Response
As I understand it, in effect this paper by and large compares aspects of what has been diagnosed as distal sensory polyneuropathy between 11 sufferers with the HIV-associated variety and 4 sufferers with other varieties of the neuropathy, however caused. If so, it seems to me that correct neurological categorisation of the disorders present is important. The diagnostic criteria described in the paper certainly indicate that the diagnostic sensory changes need to be symmetrical and distal in the lower limbs, but do not specify that their distribution be in a stocking rather than a segmental pattern, and do not consider the upper limbs where supporting evidence might be available. The bilateral reduction in ankle jerks relative to knee-jerks could be explained by lumbar-sacral disc protrusions at spinal level involving the S1 nerve roots bilaterally but sparing the L3,4 roots, with sensory change of the appropriate parts of both feet. Such disc protrusions are not rarities. Was their possible presence excluded?
Were nerve conduction studies carried out to support to the diagnostic categorisation?
Although you consider other possible causes of the peripheral nerve pathology such as diabetes, alcohol, and various non-therapeutic drugs, did you take steps to exclude other possible non-HIV causes such as familial varieties of peripheral neuropathy, industrial exposure to neurotoxins or certain therapeutic drugs that may have been prescribed in the past?
It seems to me that answers to the above questions might be helpful to readers in their assessing of your paper.
- Response:
Thank you for your valuable comment. We have not done any conductive or imaging studies, such as electromyography, intraepidermal nerve fiber density, and MRI. However, we have published previously on the correspondence of clinical evaluations of DSP and NCV [1]. In terms of the participants’ antiretroviral regimen, the frequency of DSP and its symptoms was not different between those who had prior exposure to the D-drugs (Dideoxynucleoside analog reverse transcriptase inhibitors (nRTIs) including stavudine or didanosine) and their counterparts. However, prior exposure to the D-drugs was associated with sensation loss (Table S3). We clarified the study design in Methods; updated Table S3, and provided some explanation in the discussion, copied below:
(Pages 9, Lines 301 and Page 10, Line 326):
“Participants underwent standardized evaluations of DSP conducted by centrally trained clinicians, mid-level practitioners, and physicians. Evaluations included clinical examination for neuropathy signs: bilateral symmetric distal vibratory, sharp, and touch loss in the legs and feet, and reduced ankle reflexes compared to knees. DSP was defined as having ≥2 DSP signs [36]. DSP symptoms included DNP, paresthesias, and loss of sensation. Evaluations also included assessment of DNP, defined as burning, aching, or shooting symptoms in the distal legs and feet. To exclude non-DSP-related conditions, such as mononeuropathies and radiculopathies, we mandated the presence of symmetric, bilateral, and distal abnormalities with a stocking rather than a segmental pattern when considering both signs and symptoms. Notably, we have published previously on the correspondence of clinical evaluations of DSP and nerve conductive studies [1].
Current CD4+ T-cell count, nadir CD4+ count (self-report), viral suppression, plasma and CSF HIV viral load, AIDS status, duration of HIV infection, current use, and ART duration, and prior exposure to the D-drugs (Dideoxynucleoside analog reverse tran-scriptase inhibitors (nRTIs) including stavudine or didanosine) were measured and collected for all participants. Potential confounders known to be risk factors for DSP were considered and assessed. These included a history of diabetes mellitus (self-reported or anti-diabetic medications) [42], metabolic syndrome according to AHA criteria [43], sex-specific cardiovascular disease (CVD) risk determined by the modified Framingham risk score [44, 45], as well as anthropometric and demographic characteristics, such as age, gender, ethnicity, and height [36, 42, 46]. Information regarding the history of methamphetamine and other substance abuse and dependency was gathered using the Composite International Diagnostic Interview (CIDI version 2.1) [47, 48]. This interview was structured according to DSM-IV criteria. Other Substance Use Disorders were identified as meeting specific criteria, including lifetime diagnoses of alcohol, cocaine, and opioid abuse or dependence.“
Page 8, line 233:
DSP is more common in the lower limbs than in the upper limbs. Research indicates that patients with additional causes of DSP more often experience upper limb sensory symptoms and findings, suggesting a higher prevalence of DSP in the lower ex-tremities. In the current study, we also focused on bilateral distal vibratory, sharp, and touch loss in the legs and feet, and reduced ankle reflexes compared to knees. Although the clinical diagnosis of DSP was solely clinical, and any conductive or imaging studies, such as electromyography, intraepidermal nerve fiber density, and MRI have not been done for the participants, we have published previously on the correspondence of clinical evaluations of DSP and nerve conductive studies. In addition, bilateral reflex loss (ankle jerk) is not common in most radiculopathies. However, we acknowledge that the signs might be overlapped between DSP and some radiculopathies, such as S1 radiculopathy.
Reference:
- Robinson-Papp J, Morgello S, Vaida F, Fitzsimons C, Simpson DM, Elliott KJ, et al. Association of self-reported painful symptoms with clinical and neurophysiologic signs in HIV-associated sensory neuropathy. PAIN. 2010;151(3):732-6.

Reviewer 2 Report
Comments and Suggestions for Authors
Elevated biomarkers of inflammation and vascular dysfunction are associated with distal sensory polyneuropathy in people with HIV
This is an excellent article by authors showing the association of elevated inflammation and vascular dysfunction biomarkers with distal sensory polyneuropathy (DSP) in PWH.
The present study aimed to determine if DSP signs and symptoms were associated with a panel of plasma biomarkers of inflammation (d-dimer, sTNFRII, 21 MCP-1, IL-6, IL-8, IP-10, sCD14) and vascular I integrity (ICAM-1, VCAM-1, uPAR, MMP-2, VEGF, uPAR, TIMP-1, TIMP-2) and if there is difference between PWH and people without HIV (PWoH). The findings of this study suggest that inflammation and vascular integrity alterations may contribute to DSP pathogenesis in PWH but not PWoH, plausibly through endothelial dysfunction and axonal degeneration.
The study is meticulously designed and well-executed, and the present manuscript is well-written. However, the following are the specific comments to strengthen the manuscript further,
1. The introduction mentioned that elevated levels of pro-inflammatory cytokines such as IL-1, IL-2 receptor-alpha, and tumor necrosis factor (TNF) have been associated with DSP in PWH. But why have these inflammatory markers been excluded from the present study? What is the explanation?
2. If the above-mentioned pro-inflammatory cytokines (IL-1, IL-2 receptor-alpha, and TNF) have been studied, what changes are observed between PWH and PWoH?
3. What will the consequences of ART disruption on DSP in PWH?
Author Response
This is an excellent article by authors showing the association of elevated inflammation and vascular dysfunction biomarkers with distal sensory polyneuropathy (DSP) in PWH.
The present study aimed to determine if DSP signs and symptoms were associated with a panel of plasma biomarkers of inflammation (d-dimer, sTNFRII, 21 MCP-1, IL-6, IL-8, IP-10, sCD14) and vascular I integrity (ICAM-1, VCAM-1, uPAR, MMP-2, VEGF, uPAR, TIMP-1, TIMP-2) and if there is difference between PWH and people without HIV (PWoH). The findings of this study suggest that inflammation and vascular integrity alterations may contribute to DSP pathogenesis in PWH but not PWoH, plausibly through endothelial dysfunction and axonal degeneration.
The study is meticulously designed and well-executed, and the present manuscript is well-written. However, the following are the specific comments to strengthen the manuscript further,
- The introduction mentioned that elevated levels of pro-inflammatory cytokines such as IL-1, IL-2 receptor-alpha, and tumor necrosis factor (TNF) have been associated with DSP in PWH. But why have these inflammatory markers been excluded from the present study? What is the explanation?
- If the above-mentioned pro-inflammatory cytokines (IL-1, IL-2 receptor-alpha, and TNF) have been studied, what changes are observed between PWH and PWoH?
- What will the consequences of ART disruption on DSP in PWH?
- We thank the reviewer for this insightful comment on excluding certain pro-inflammatory cytokines from our study. The selection of biomarkers was guided by our research focus and data availability. Unfortunately, we did not measure IL-1, IL-2 receptor-alpha, and tumor necrosis factor (TNF) in this set of samples. However, we are going to figure out the association between these inflammatory biomarkers and other plasma and cerebrospinal fluid (CSF) biomarkers with DSP in our future projects and we acknowledge the significance of the excluded cytokines in the context of DSP pathogenesis in PWH. We have discussed this limitation in the revised manuscript and proposed the inclusion of these cytokines in future research to provide a more comprehensive understanding of the inflammatory processes involved in DSP among PWH.
- In terms of ART disruption, we thank the reviewer for raising the question regarding the consequences of ART disruption on DSP in PWH. We have included a discussion in the revised manuscript that contextualizes our findings within the existing literature, emphasizing the importance of continuous ART in managing DSP symptoms in PWH. This addition not only clarifies the implications of our study but also underscores its contribution to ongoing discussions on ART management and DSP in the context of HIV. We added the following explanation regarding the consequences of ART disruption on DSP in PWH in the discussion:
Page 8, Line 221: Interruptions in ART for PWH can lead to adverse outcomes, such as worsening of DSP symptoms and an increased risk of HIV disease progression due to decreased CD4 counts and nadir CD4 counts. Specifically, PWH who initiated CART after their CD4 counts dropped below 350 cells/μL were found to have a significantly higher prevalence of DSP compared to those who started CART before their CD4 counts fell below this threshold.
Moreover, some studies suggest that ART itself may be neurotoxic, as it has been associated with a higher prevalence of DSP even after controlling for other factors, including exposure to D-drugs. However, further research is needed to fully understand the relationship between ART and DSP in PWH. The current study found that 87% of PWH were receiving ART, and no correlation was observed between the duration of ART and the presence or severity of DSP. However, a history of exposure to D-drugs was associated with an increased risk of sensory loss in PWH.
Page 8, Line 251: Besides, we did not measure IL-1, IL-2 receptor-alpha, and tumor necrosis factor (TNF), the biomarkers that have been previously observed to be linked with DSP, in this set of samples. We acknowledge the significance of these excluded cytokines in the context of DSP pathogenesis in PWH W and recommend designing a study to figure out the association between these inflammatory biomarkers and other plasma/CSF biomarkers with DSP in future projects.

Round 2
Reviewer 1 Report
Comments and Suggestions for Authors
I have no further neurological matters to raise.